# Effects of Carbon Nanomaterials and *Aloe vera* on Melanomas—Where Are We? Recent Updates

**DOI:** 10.3390/pharmaceutics14102004

**Published:** 2022-09-22

**Authors:** Elidamar Nunes de Carvalho Lima, Guilherme Leão Barros Martins, Ricardo Sobhie Diaz, Mauro Schechter, José Roberto Castilho Piqueira, João Francisco Justo

**Affiliations:** 1Telecommunication and Control Engineering Department, Polytechnic School of the University of São Paulo, Avenida Prof. Luciano Gualberto, Travessa 3, 158, São Paulo 05508-010, Brazil; 2Infectious Diseases Division, Department of Medicine, Federal University of São Paulo, São Paulo 04023-062, Brazil; 3Electronic Systems Engineering Department, Polytechnic School of the University of São Paulo, São Paulo 05508-010, Brazil

**Keywords:** melanoma, tumor, cancer, *Aloe vera* compounds, carbon-based nanomaterials, functionalization, antineoplastic, immunotherapy

## Abstract

Melanoma is an aggressive skin cancer that affects approximately 140,000 people worldwide each year, with a high fatality rate. Available treatment modalities show limited efficacy in more severe cases. Hence, the search for new treatment modalities, including immunotherapies, for curing, mitigating, and/or preventing cancer is important and urgently needed. Carbon nanoparticles associated with some plant materials, such as *Aloe vera*, have shown appealing antineoplastic activity, derived mainly from the compounds aloin, aloe-emodin, barbaloin acemannan, and octapeptide, thus representing new possibilities as antitumor agents. This systematic review aims to arouse interest and present the possibilities of using *Aloe vera* combined with carbon-based nanomaterials as an antineoplastic agent in the treatment and prevention of melanoma. Limitations and advances in melanoma treatment using functionalized carbon nanomaterials are discussed here. Moreover, this review provides the basis for further studies designed to fully explore the potential of carbon nanomaterials associated with *Aloe vera* in the treatment of various cancers, with a focus on melanoma.

## 1. Introduction

Cancer is a global health issue, as it is one of the most frequent diseases affecting human populations and a significant cause of death worldwide. Melanoma is an aggressive skin cancer that affects approximately 3% of the global population and is responsible for more than 80% of deaths from skin cancer [1]. In the United States, the number of melanoma cases has increased substantially in recent decades, with an incidence of 300/1,000,000 population and an increase in the frequency of more than 300% between 1994 and 2014 [2]. Skin cancers are diagnosed in approximately 200,000 Americans every year, and 1,000,000 U.S. citizens live with melanoma, making this one of the most common skin cancers in the United States, as well as worldwide [3]. Therefore, melanoma is a global health issue with psychosocial consequences, and the development of new technologies for its effective treatment deserves increasing attention.

Melanoma results from the malignant transformation of melanocytes, cells derived from the neural crest. Although usually located in the skin, melanomas can develop in other parts of the body where neural crest cells migrate, such as the brain and the gastrointestinal tract. It predominantly affects adults, with no sex preference. The risk of developing a second melanoma after the diagnosis of a first melanoma is 3–5% [4]. Risk factors for developing melanoma include a family history (an affected relative doubles the risk) and personal characteristics, such as blue eyes, red hair, pale complexion, ease of sunburning, freckling, and melanocytic nevi (the number is more relevant than size) [5]. High levels of ultraviolet radiation (UVA and UVB) exposure are associated with the development of melanomas. Sunscreens mainly block UVB, and people who constantly use them may be more exposed to UVA than the general population. Interestingly, some data indicate that the risk of melanoma is higher in people who use sunblock. A low socioeconomic status is associated with more advanced diseases [1,2].

The most common occurrence is characterized by skin lesions with a recent change in color, size or shape. A mnemonic aid commonly used to recognize a melanoma is ABCDE: A for asymmetry, B for irregular borders, C for color variation (especially red, white and blue tones in a brown or black lesion), D for a diameter greater than 6 mm, and E for an elevated surface. A definitive diagnosis is based on histopathology of biopsies of lesions. The four main histological classifications of melanoma are superficial spreading of melanoma, nodular melanoma, lentigo and acral lentiginous melanoma. Other histological types represent a small fraction of cases. Superficial spreading of melanomas represent 70% of all cases. Several classification schemes have been proposed, and the two most commonly used classifications are based on the vertical thickness of the lesion or the anatomic level of invasion of skin layers [6]. Additionally, several staging systems have been proposed based on the involvement of the dermis, thickness, and the involvement of local, regional, or distant lymph nodes or organs [1,3].

Lesion excision is the primary form of treatment. Drug treatment is used as an adjuvant in patients with lesions that cannot be resected, patients with metastases, and patients with resected advanced disease. Several monoclonal antibodies and humanized monoclonal antibodies have been approved by the U.S. Food and Drug Administration (FDA) for clinical use [7,8]. A modified, live attenuated herpes simplex virus programmed to replicate within tumors and manufacture granulocyte–macrophage colony-stimulating factor (GMCSF) was recently approved by the FDA for the local treatment of some unresectable tumors. The prognosis mainly depends on the disease stage. Five-year survival rates range from 97% when diagnosed in the earliest stages of the disease to 0% when the diagnosis is made at the most advanced stages of the disease. Therefore, although several therapeutic options are available to treat melanomas, including surgery, radiation therapy, and chemotherapy, much room for improvement exists [9,10]. 

The lack of cell specificity in treatment often leads to adverse events due to harmful effects on healthy cells [8,9]. Numerous attempts to circumvent this problem have been investigated, including the use of plants with medicinal potential. Phytotherapy is used for antitumor treatment by a significant portion of the population [11]. A potentially effective approach to fight different types of tumors is the use of functionalized nanoparticles as “molecular chassis” or drug carriers. These drug delivery systems (DDSs), called therapeutic nanoplatforms, allow the coupling of multiple drugs [12]. Nanoparticles, as parts of a DDS, increase bioavailability and specificity, thus improving therapeutic efficacy by increasing drug retention in the tumor microenvironment. Additionally, nanoparticles facilitate the maintenance of continuous and tunable drug release in the target region [13]. 

In this regard, nanoparticle systems represent the Holy Grail in many applications, not only in objective/specific drug delivery by DDSs but also in tumor targeting, diagnostics, cellular imaging, and image-directed tumor ablation with excellent theranostic action [14]. In this context, nanoparticles penetrate the skin, reach the tumor microenvironment, and facilitate targeted drug delivery [12]. Thus, extensive research efforts have been employed in recent years to develop and functionalize novel and promising nanoparticles into unified methods for diagnosis and therapy, i.e., in theranostics, to treat different types of cancer, including skin cancer [12,15].

Several nanoformulations are already part of clinical practice, such as doxorubicin-containing PEGylated liposomes named Doxil^®^, which are used for the treatment of ovarian and breast cancer, Kaposi’s sarcoma, and multiple myeloma [16]. Abraxane^®^ is also an antineoplastic treatment used in clinical practice and is composed of paclitaxel nanoparticles linked to albumin, which are used to treat different types of cancer, including lung and breast cancer [17]. All these nanoformulations have been approved by the FDA since 2005 [18]. Regarding skin cancer, a large number of nanostructured materials, including nanotubes, quantum dots, liposomes, dendrimers, nanomicelles, polymersomes, gold nanoparticles (NPs), nanogels, silica NPs, polymeric NPs, nanospheres, magnetic NPs, nanostructured lipid carriers, and solid lipid NPs, are under development [19,20]. 

Several types of nanomaterials have been investigated for the treatment of skin diseases and melanomas. These nanoformulations include dendrimers, liposomes, carbon derivatives, and protein-based and inorganic nanoformulations. Thus, the implementation of NPs in antineoplastic therapeutics provides targeted NP/drug delivery to melanoma, with increasing prospects related to the specificity and therapeutic efficiency [12]. Among the numerous advantages of nanomedicine, a striking feature is the possibility of binding the polymer to a malignant cell membrane or specific cytoplasmic sites through interactions with nuclear receptors, thereby optimizing the concentration of these therapeutic biomolecules at the target site and reducing their toxicity [19]. In this sense, nanosized particles increase the therapeutic specificity, with better biopermeability of the compounds, thereby reducing adverse reactions [19,21,22].

Here, we summarize, describe, and schematize the most relevant findings related to the use of *Aloe vera* associated with carbon NPs as phytotherapeutics with antineoplastic applications by focusing on melanoma, and we present and discuss the main challenges and advances. Thus, we present an updated and comprehensive review of the combination of nanomaterials with *Aloe vera* for the treatment of melanoma, providing insights into the most frequently used and varying types of carbon nanomaterials with antitumor activity and continuing the studies of our research group on carbon-based nanomaterials with potential immunotherapeutic/antineoplastic bioapplications [23,24].


**Parameters considered in this review—Identification of relevant studies and research**


We systematically searched the literature using the following electronic databases to develop this study: PubMed-NCBI, MEDLINE-Bireme, Google Scholar, and ScienceDirect. Thus, this review was conducted by considering the main papers published over the last 27 years. The main scientific articles were selected using relevant keywords such as “carbon-based nanomaterials” OR “melanoma” OR “*Aloe vera*” OR “immunotherapy/phytotherapy” OR “nanoparticles/functionalization” OR “antitumor nanomedicine”. Inclusion criteria considered when selecting the literature for this study included (i) an initial literature review of articles containing the exact keywords; (ii) a subsequent literature review containing the associated words in the selected articles, and (iii) reviews published in English. As exclusion criteria for this study, we did not consider articles that (i) were not written in English or (ii) did not contain the exact and/or associated words cited in the title or abstract of the reference article. Two reviewers screened the articles independently. A total of 157 articles were included in this final review based on the relevance of the results and the use of nanotechnologies, especially carbon-based nanomaterials, as antitumor agents against melanoma. Thus, our work considered clinical case articles, review articles, prospective studies focusing on immunotherapy/phytotherapy for melanoma, and the use of carbon-based nanomaterials as a theranostic and/or antitumor-antineoplastic agent.

## 2. Genetic Basis of Melanoma

The maintenance of order in a multicellular organism requires extremely precise and punctual controls, including the correction of errors during cell multiplication, alterations in the inhibition of cell proliferation or apoptosis, and modification of the maintenance of the intracellular environment [25]. Accordingly, the breakdown of intracellular homeostasis that may occur, for example, due to the accumulation of genetic mutations, dysregulates the mechanisms of cell proliferation and disrupts the homeostasis of the organism as a whole, which may give rise to cancer [26].

Several mechanisms are involved in the processes of tumor growth and progression, including invasive growth, immune evasion, replicative immortality, and inflammation [26]. All these factors are related to cell invasion, metastasis, genomic instability, and mutation, which deregulate signaling pathways, including cell proliferation and apoptosis. In terms of genetic factors related to the development of melanomas, several metabolic pathways with genetic modulations are involved in the processes of tumorigenesis, including mutations in the BRAF gene type V600E, which encodes a serine/threonine-protein kinase involved in intracellular signaling related to cell growth [3,4,27,28]. 

In addition to mutations in some genes, the penetrance of certain alleles contributes to the risk of melanomas, as they affect and/or amplify factors related to failures during the process of checking the transition of cell replication, specifically during the transition from the G1 to S phase, or failures in the action of telomerase on chromosomes [27]. In this regard, the main genes influenced by the presence of certain alleles include SLC45A2, TYR, MC1R, OCA2, and ASIP, which are all directly or indirectly related to the process of skin and hair pigmentation. Thus, the main genes related to the process of melanoma tumorigenesis but not associated with the process of skin and hair pigmentation, as well as the type of mutation, are shown in Table 1.

## 3. Phytotherapeutic Prospects for *Aloe vera*

Phytotherapy, which is defined as the use of plants containing medicinal compounds, consists of the use of products or subproducts of plant origin to solve therapeutic problems related to the prevention, treatment, and/or cure of a pathological state. These drugs are based on molecules present in various plant species with various biological activities and purposes. These biologically active molecules may be products of the primary or secondary metabolism of plants whose pharmacological activity is achieved or maximized under certain laboratory conditions [34,35]. In this context, phytotherapeutic compounds have attracted the interest of the scientific community and the pharmaceutical and dermo-cosmetic industries, enabling regenerative processes that are fundamental for both the maintenance and prolongation of the life of humans [36].

Natural products are important resources that have been used for many years in traditional medicine for the prevention and treatment of numerous pathologies. In this context, *Aloe vera*, which has been cited in the literature for thousands of years, has initial records of its applications beginning 2100 B.C. in Mesopotamia, exerts antioxidant effects and has become known as the “plant of immortality” [37]. *Aloe vera* is an herbaceous plant belonging to the Aloaceae family, which contains approximately 15 genera and 420 species identified worldwide that are easy to cultivate and adapt readily to soils with little water or poor nutrients [38]. Popularly, it is known as *Aloe barbadensis* Miller, *Aloe vera* Linne, and commonly *Aloe vera* or simply babosa. *Aloe vera* has green, thick, succulent foliage ranging in size from 30 to 60 cm that takes 4–5 years to reach maturity and become usable [39]. It is a plant used in many countries and various cultures with different medical traditions, such as Ayurveda, Siddha, Unani, and homeopathy [40].

Studies have documented the role of *Aloe vera* in the prevention and treatment of several pathologies, which is related to its approximately 75 active compounds with high power and therapeutic value (Figure 1) [38]. The different compounds that are present in *Aloe vera* have properties mediating important biological functions, including but not limited to anti-inflammatory, healing, antioxidant, antibacterial, antimicrobial, antifungal, and antineoplastic effects [37,39]. Thus, the anticancer activities of *Aloe vera* may be useful and efficient for the treatment or prevention of various types of cancer, including melanomas [38]. The components present in *Aloe vera*, including anthraquinones and aloe-emodin, inhibit the proliferation of breast cancer, cervical cancer, and hepatocellular cancer cell lines. Accordingly, the use of phytotherapy based on *Aloe vera* components as an alternative and/or complement to antineoplastic treatment represents an important immunotherapeutic strategy [38]. 

The best phytotherapeutic properties observed in *Aloe vera* are attributable to the presence of compounds such as acemannan, proteins, glycoproteins, aloin, aloe-emodin, anthraquinones, alloins, barbaloin, isobarbaloin, polysaccharides, vitamins A, B, C, and E, amino acids, enzymes, carbohydrates and elements such as calcium, potassium, magnesium, and zinc (Figure 1) [38,41]. Table 2 presents some components of *Aloe vera* that are of major scientific interest and their characteristics. Among the main phytochemicals, phenolic compounds such as alkaloids and flavonoids have antimutagenic and antitumor actions. In addition to these compounds, aloe-emodin and aleosin possess antimicrobial, anti-inflammatory, and antimutagenic activities [37]. Aloe-emodin (AE, 1,8-dihydroxy-3-hydroxymethylanthraquinone) is an anthraquinone component present in *Aloe vera* with antiproliferative and proapoptotic effects on different tumor cells, such as lung, liver, and colon cancer cells. These compounds activate molecular mechanisms that induce apoptosis, disrupt the cell cycle, alter metastasis, and improve immune functions. In addition, AE acts as a sensitizer for chemotherapy and radiation in antitumor therapy [42].

### 3.1. Healing Activities

*Aloe vera* exhibits cellular regenerative activity, which is linked to the modulation of different stages of cellular regeneration, with phytotherapeutic effects that amplify and/or improve the efficiency of the stages that compose the process of cellular/tissue regeneration [38]. Studies have reported the antioxidant activities of *Aloe vera* provided by anthraquinones and anthrones, while the compound aleosin has antioxidant activity and inhibits important enzymes in the regenerative process, including the enzymes COX-2 and thromboxane A2 synthetase [43,44].

The substances present in *Aloe vera* that exhibit cicatrizing/regenerative activities of commercial/pharmaceutical interest include glycoproteins [38,39]. Significant increases in the proliferation of gingival fibroblasts and stimulation of the secretion of keratinocyte growth factor-1 (KFG-1) and vascular endothelial growth factor (VEGF), substances involved in cellular regeneration and healing, were observed when *Aloe vera* glycoproteins were added to collagen type I. In addition, *Aloe vera* stimulates macrophages to release interleukin-6, nitric oxide, and tumor necrosis factor-alpha (Figure 2a) [45,46]. 

**Table 2 pharmaceutics-14-02004-t002:** Main compounds present in *Aloe vera* and their multiple applications.

Compound Type	Substance	Chemical Formula	Activities [Reference]
Carbohydrates	Acemannan	C_66_H_100_NO_49_	Skin protection, antimicrobial activity, prebiotics, protection against digestive diseases, and immunomodulation [45]
Anthraquinones/ Anthrones	Aloe-emodin	C_15_H_10_O_5_	Antidiabetic effects, antimelanoma, bone protective, cardioprotective, antitumor, antimicrobial, and prebiotic activities [47]
Anthraquinones/ Anthrones	Aloin	C_21_H_22_O_9_	Bone protection, skin protection, digestive system, antidiabetic effects, antitumor effects, anti-inflammatory effects, and laxative/purgative effects [48]
Chromones	Aleosin	C_19_H_22_O_9_	Skin protection and antitumor effects [45]
Anthraquinones/ Anthrones	Emodin	C_15_H_10_O_5_	Antitumor, antibacterial, antifungal, antiviral, anti-inflammatory, antiulcer, and diuretic effects [47]
Anthracenes	Barbaloin	C_21_H_22_O_9_	Laxative and purgative effects [48]
Glycoproteins	Octapeptide	C_35_H_61_N_13_O_10_	Antitumor effects [49]

### 3.2. Photoprotective Activities

Premature aging or photoaging of the skin, which is characterized by wrinkles, a leathery texture, and mottled pigmentation, is a direct consequence of exposure to sunlight. UVA rays are a major risk factor for cancer and are associated with the induction of inflammation, immunosuppression, photoaging, and melanogenesis [3,4]. However, studies have documented the photoprotective effects of *Aloe vera* (*Aloe barbadensis*) on UVB radiation in cells by conducting both morphological and numerical observations, especially among accessory cells, epidermal dendritic cells, and Langerhans cells [50,51]. The photoprotection conferred by *Aloe vera* is associated with the maintenance of membrane integrity, as well as in intracellular organelles, with increased lysosomal stability, which decreases lipofuscinogenesis and consequently cell death [52].

An attenuation of lysosomal damage and inhibition of autophagy of in human immortalized skin keratinocyte (HaCaT) have been observed when compounds from *Aloe vera* were used. Thus, *Aloe vera* decreased the redox imbalance induced by both UVA radiation and MB photoactivation at λ = 633 nm [51]. *Aloe vera* binds and assembles phospholipid groups, preventing cell membrane leakage and decreasing the exposure of cell membrane reactive sites [37]. Accordingly, cytoprotection is related to membrane protection through a mechanism similar to that induced by trioses. Other studies have shown that compounds from *Aloe vera* suppress reactive oxygen species (ROS) production, which is induced by UVA, resulting in decreased intracellular lipid peroxidation, an increased endogenous antioxidant capacity, and increased cellular capacity/viability [51]. In addition, studies have shown a reduction in UVA-induced photodamage in lysosomal membranes by natural compounds, such as flavonoids, which protect against lysosomal destruction caused by UVA radiation, the photooxidation of lipid membranes, and damage to lysosomal membranes (Figure 2b) [52]. Even upon exposure to visible light, *Aloe vera* extract significantly protects human keratinocytes from photodamage caused by solar radiation [51].

### 3.3. Antiviral Activities

Currently, synthetic antiviral drugs and methods, including nucleic acid protein inhibitors, neuraminidase inhibitors, ion channel blockers, and siRNA techniques, have some limitations in their use, including the emergence of resistant strains, high therapeutic costs, and harmful side effects of medications [53]. In contrast to all these treatments, phytotherapeutics have low cytotoxicity and cost and multidirectional effects because they act not only as antiviral agents but also exhibit an excellent ability to stimulate the immune system to combat viral infections [36]. The classically defined antiviral mechanisms of medicinal plants are related to the inhibition of viral replication, blocking the attachment or entry of the virus into host cells, and the inactivation of the virus, which prevents viral infection. Furthermore, a constant interest in using herbal medicines has been noted, especially due to the preference for natural medicines and constant concerns about the toxic effects of synthetic materials [11,54].

Sonicated phytotherapeutic *Aloe vera* shows higher antiviral activity than nonsonicated *Aloe vera* extracts. The IC50 of sonicated *Aloe vera* (AV-WNS) is 77.16 mg/mL, compared to the IC50 values of AV-W25, AV-W50, and AV-W100, which are 13.01 mg/mL, 9.69 mg/mL and 4.94 mg/mL, respectively [55]. Reducing the IC50 improves the antiviral activity of the herbal extract. In this context, the highest antiviral activity against the influenza virus was observed for AV-W100, without evident cytotoxicity. The influenza A virus H7N9 and the H5N1 virus, which are highly pathogenic, pose challenges to public health [56]. Studies have shown that anthraquinone derivatives, such as aloe-emodin, emodin, and chrysophanol, exhibit antiviral activity [47]. Thus, previous records have shown inhibitory effects os *Aloe vera* (extracted in 2% dimethyl sulfoxide (DMSO) on the herpes simplex virus in Vero cell lines ranging from 0.2–5% (Figure 2c). 

*Aloe vera* is an herbal medicine, has been reported to inhibit viruses such as human cytomegalovirus, herpes simplex virus type 2 (HSV-2), and poliovirus [37,57]. Thus, *Aloe vera* shows significant anti-influenza activities; in particular, the bioactive compounds of this plant extracted using ultrasound exhibit unprecedented activity against influenza [55]. Proteomic analyses using MDCK cells indicated that galectin-emodin acts on galec-threonine-3, thioredoxin, and nucleoside diphosphate kinase A, where galection-emodin and aloe-emodin induce galectin-3 overexpression [57,58]. In addition, recombinant galectin-3 increases the expression of antiviral genes, such as IFN-β, IFN-γ, PKR, and 2′,5-OAS, in infected cells. Galectin-3 inhibits influenza virus replication and exhibits cytokine-like regulatory actions via JAK/STAT signaling pathways. Aloe-emodin was observed to restore STAT1-mediated antiviral responses inhibited by NS1 in infected cells [55]. Thus, aloe-emodin is a major component contributing to the inactivation of influenza viruses, with promising results as a potent antiviral therapeutic [47,58]. 

The emodin compounds from *Aloe vera* affect both DNA and RNA viruses by acting on the viral envelope, preventing the adsorption of the virus and thus its further replication, but do not affect viruses in the extracellular medium [47,57]. One of the mechanisms by which acts as an antiviral against Japanese encephalitis viruses, enteroviruses, adenoviruses, and rhinoviruses is related to IFN signaling pathways. Additionally, increasing the aloe-emodin content in the experiments results in a directly proportional increase in antiviral activity [55,56]. Accordingly, aloe-emodin is a key inhibitor of viral entry and subsequent replication, interfering with the casein kinase 2 (CK2), Nrf2, TLR4, P38/JNK, NF-kB, galectin-3, STAT1, and INF signaling pathways (Figure 2c) [56,58]. The disruption of one, multiple, or all of these pathways is critical for inhibiting the attachment or entry of the virus into host cells, the synthesis of proteins, mRNA, and other nucleic acids, and the assembly and release of viral particles, all of which lead to inhibition of the viral replicative cycle [55].

Furthermore, the efficacious antiviral activity of *Aloe vera* against the HSV-2 virus was detected in Vero cell lines, not only during the processes of virus attachment and entry but also in the stages after viral entry, including replication [55]. The IC50 in Vero cells before virus entry was 428 µg/mL, and the CC50 value corresponding to cytotoxicity was 3238 µg/mL. Additionally, the IC50 of *Aloe vera* extract in the early stages of postfixation viral replication was 536 µg/mL, and the SI value for inhibition of the postfixation stages of HSV-2 replication was 6.04. Therefore, natural *Aloe vera* compounds represent good candidates for the development of antiviral herbal drugs [41,55,56].

### 3.4. DNA and Telomerase Activities in Tumor Cells

The compound aloe-emodin induces DNA damage in lung cancer cells by promoting ROS generation [47]. Similar effects were observed on breast, leukemia, colon, and glioblastoma multiforme cancer cells. The DNA damage in these cancer cells is maintained by aloe-emodin, which inhibits DNA repair processes [59]. 

Similarly, telomerase, an enzyme that controls telomere synthesis, is activated in many types of tumors, activating the replication, proliferation, and metastasis of these cancer cells [60,61]. One issue is that the formation of G-quadruplex structures may inhibit the activity of these enzymes by blocking the telomeric substrate in an inactive conformation, which will not be recognized or elongated by telomerase [62]. Anthraquinone was one of the first ligands found to stabilize these G-quadruplex structures, thereby inhibiting telomerase activity (Figure 2d) [59]. In addition, the compound aloe-emodin, derived from aloe-emodin 3 (AED3), an anthraquinone, has the potential to exert the same effect [63]. 

The compounds emodin, aloe-emodin, and AED3 induce a temporal increase in the fluorescence of 12C5TG-AgNC, indicating that they are G-quadruplex interacting ligands [64]. In vivo studies using murine models have shown that the compound di-2-ethylhexyl phthalate decreases telomerase activity and increases TNF synthesis [65]. According to previous studies, aloe-emodin is a competitive telomerase inhibitor and stabilizes the G-quadruplex structure in breast cancer cells. These compounds decrease telomerase activity by competing with dNTPs for binding to the enzyme active site, stabilizing the telomeric G-quadruplex structure [59,62].

### 3.5. Immunomodulatory Activities

*Aloe vera* compounds exhibit immunomodulatory activities by potentiating the function of the immune system in the antitumor response [48,66]. Lee et al. pioneered studies reporting the immunomodulatory effects of *Aloe vera* [67]. *Aloe vera* is one of the most widely used herbal medicines in natural treatment and as an alternative therapy for various types of inflammatory and infectious diseases. One of the mechanisms of action of *Aloe vera*, modulation of the immune response, occurs through the activation of macrophages, which play an essential role as the first line of defense against invading pathogens [37,68]. Additionally, after administration of *Aloe vera* to a murine model (400 mg/kg, orally), the acemannan compounds present in *Aloe vera* stimulated the production of immunomodulatory substances from macrophages, including cytokines, nitric oxide, and the expression of surface molecules, and cellular morphological changes, with improvements in humoral/secondary immunity [69]. 

Randomized studies of patients with metastatic stage cancer have compared chemotherapy to chemotherapy in combination with *Aloe arborescens* and have shown a significant increase in the number of lymphocytic cells in patients treated concomitantly with aloe compounds compared to patients treated with chemotherapy alone [70]. Additionally, aloe-emodin increased the levels of interleukin (IL)-1beta and tumor necrosis factor (TNF)-alpha, and all the effects observed during treatment with these herbal compounds have been described in both in vivo and in vitro evaluations of the immune system mediated by murine and human lymphoid cells [46,62].

Aloe-emodin is involved in inhibiting fibrosarcoma growth but shows no cytotoxicity toward normal human cells. Lectin, another compound derived from aloe, exerts a cytotoxic action on the surface of the tumor cells, with increased tumor specificity and efficient immune activation of T cells [43]. Another important factor is that acemannan, one of the most active polysaccharides found in *Aloe vera*, and it is known to exert antitumor activity. These mechanisms are mediated by the activation of macrophages and the release of TNF, interleukin-1, and interferon [37,45]. In addition, acemannan participates in enhancing immunity in radiation-damaged cells and tissues. An intraperitoneal injection of acemannan stimulates the synthesis of monokines, which induces immune activation through the recruitment of cells by interleukin-1 and TNF and the regression of tumors (Figure 2e) [45].

### 3.6. Anti-Inflammatory and Antimetastatic/Antiproliferative Activities

Inflammation is linked to several processes involved in tumorigenesis through the delivery of bioactive molecules to the tumor microenvironment, including growth factors that sustain the cell proliferative signal, survival factors that limit cell death, proangiogenic factors, extracellular matrix-modifying enzymes, which promote angiogenesis, invasion, and metastasis, and signaling pathways that stimulate the deleterious epithelial-mesenchymal transition (EMT) [60,71]. In addition, inflammatory cells release substances that are actively mutagenic to cancer cells, such as ROS [59]. On the other hand, several studies have documented the anti-inflammatory activity of aloe compounds in inhibiting edema [37,69].

Previous results have shown that aloe-emodin suppresses metastasis in breast cancer cells, and although these mechanisms have not been fully clarified, they seem to be related to the inhibition of invasion and migration of these tumor cells [65]. Another possibility is that these antimetastatic mechanisms might be related to a decrease in the levels of proteins responsible for tumor metastasis, suppressing cell proliferation in tumors such as neuroblastomas [66,68]. In addition, the compound aloe-emodin inhibits key regulatory molecules involved in colon cancer cell migration. The same antiproliferative effect has been observed on other cancers, including carcinomas, melanomas, gastric cancer, and hepatocellular carcinoma, with a significant decrease in cell migration [37,65].

Previous in vitro and in vivo studies have shown the antineoplastic activity of *Aloe vera* that is related to at least three distinct mechanisms: (i) antiproliferative (ii) immunostimulatory, and (iii) antioxidant mechanisms [37,72,73]. These antiproliferative activities of *Aloe vera* are mainly mediated by anthraquinone molecules, such as aloe-emodin, aleosin, and aloin, also known as barbaloin. The immunostimulatory activity, on the other hand, is mainly related to the compounds acemannan and aloemannan, both of which exert antitumor activity by stimulating immune responses (Figure 2f) [45]. Thus, according to previous studies, the main compounds used in the treatment of skin cancer comprise aloe-emodin, barbaloin, aleosin, octapeptide, and acemannan. All these antitumor compounds are derivatives of hydroanthraquinones, which have carbonic rings in their structure and are attached mostly to polar groups, such as oxygen and hydroxyls [47,69].

The compounds in *Aloe vera* have potent activity against several types of cancers, including Hodgkin’s and non-Hodgkin’s lymphoma, cervical carcinoma, osteosarcoma, malignant glioma, chronic leukemia, Merkel cell carcinoma, leukemia, lymphoma, pancreatic cancer, myeloma, sarcoma, hepatoma, neuroblastoma, melanoma, lung carcinoma, glioblastoma multiform, fibrosarcoma, ocular surface squamous neoplasia, promyelocytic leukemia, breast, soft tissue, lung, acute, bladder, ovarian, gastric, colon, oral, colorectal, prostate, nasopharyngeal, bladder, hepatocellular, duodenal, neuroectodermal, liver, and pancreatic cancers, as well as liver metastasis of pancreatic cancer and acute myeloid leukemia, among others [37,65,68]. The compounds in *Aloe vera* act in the cell replicative cycle specifically during the sub-G0, sub-G1, and S phases, which entails prolonging the replicative duration of these cells, inducing apoptosis [68]. Another effect is the modulation of genes responsible for replication and cell cycle control, such as cyclin D1, Bax, CYP 1A1, CYP 1A2, AKT, ERK 1/2, p53, and p21 (Figure 2f). Modulation of tumor cell characteristics, as well as synergistic actions with other compounds, such as cisplatin, which is commonly used to treat various tumors, has also been observed in vitro [59,64,74].

Previous results from both in vitro and in vivo studies have described the optimization of the antitumoral activity of *Aloe vera* compounds when used together with chemotherapy, potentiating antitumoral treatment [37,54]. Thus, in addition to the antiproliferative, immunostimulatory, and antioxidant activities, the compounds present in *Aloe vera* may act synergistically with existing drugs and therapies for the prevention and cure of tumors, which would be essential for the efficacy and robustness of antineoplastic treatment, providing excellent immunotherapeutic possibilities. 

The antiproliferative effect on tumor cells is due to a series of organic actions, including disturbances in the cell cycle and differentiation of tumor cells, stimulation of the immune system, and antioxidant activities [68]. *Aloe vera* reduces not only the proliferation of tumors/metastasis but also the tumor size/mass. Thus, patients receiving chemotherapeutic treatment and/or those with metastatic tumors who were treated with *Aloe vera* in combination with chemotherapeutic treatment experienced a regression of the tumor and symptoms caused by chemotherapy, such as fatigue and asthenia and, consequently, experienced prolonged survival when treated with the associated therapy [48,75]. 

A remarkable characteristic of *Aloe vera* is its antiproliferative activity toward cancer cells, and this effect was observed in murine hepatoma pleural tumors. The dichloromethane (CH_2_Cl_2_) extract of *Aloe vera* inhibited the growth of Ehrlich ascites cells by decreasing DNA synthesis and inducing the accumulation of cells in the G1 phase of the cell replicative cycle [46,59]. In addition, *Aloe vera* exerts a cytotoxic action on tumor cells, for example, as observed with the compound emodin, a natural anthraquinone present in *Aloe vera* that is directly involved in cytotoxic activities in human myelomas [47]. Additionally, another compound, aloin, a natural anthracycline, is similar to anthracycline class drugs, such as doxorubicin, which are very commonly used in clinical practice in the treatment of several types of cancer, including breast, lung, bladder, ovarian, gastric, osteosarcoma, soft tissue cancer, Hodgkin and non-Hodgkin lymphoma, solid tumors in children, acute cancer, and chronic leukemia [16,73]. 

Compared to the control group, myeloma cells treated with *A. arborescens* exhibited a significant reduction in their proliferative capacity. Aloe-emodin inhibits the proliferation of different types of cancer cells through several mechanisms, culminating in apoptosis [47,63]. Antiproliferative effects have also been observed on colon cancer cells, oral squamous cell carcinoma, gastric cancer, colorectal cancer, cervical cancer cells, lung squamous cell carcinoma, malignant glioma, prostate cancer, nasopharyngeal carcinoma, bladder cancer, and hepatocellular carcinoma cells (Figure 2f) [59,64].

Other studies have reported the antiproliferative activities of aloe-emodin in Merkel carcinoma cells, as well as its antineuroectodermal activity. The antiproliferative and antihepatocarcinogenic effects of *Aloe vera* are mediated by the modulation of apoptotic pathways. Furthermore, aloe-emodin has been reported to exert antiproliferative/anticancer effects on both multidrug-resistant leukemia cells and lymphoma cells [47,65,68].

Aloin, another compound from *Aloe vera*, possesses antiproliferative/antitumor activities that promote apoptotic mechanisms in cervical carcinoma cells and gastric cancer cells [68,73]. Emodin also exerts antitumor activity against pancreatic cancer cells through mechanisms that promote a decreased mitochondrial membrane potential. Thus, the emodin compound azide methyl anthraquinone derivative (AMAD) blocks the phosphorylation of Her2/neu, suppressing the growth, transformation, and metastasis of cancer cells, which has similar effects to standard cytotoxic therapeutic agents and thus may represent a potential therapeutic strategy for blocking signaling pathways [69,70]. Similar results have been observed in hepatoma and ovarian cancer cells, suggesting that allomycin inhibits both the growth and proliferation of these cancer cells by interrupting the cell cycle and activating apoptosis. The compound di(2-ethylhexyl) phthalate (DEHP) from *Aloe vera* exerts antileukemic, antimutagenic, and apoptotic effects [68,76]. 

Among the main advances in phytotherapy with *Aloe vera*, antineoplastic activity against several types of cancer, including melanoma, has shown positive results, related both to tumor regression and an improvement in patients’ quality of life with reduced chemotherapy symptoms [38,48]. Murine models with transplanted tumors display a reduction in tumor mass and a significant extension of the life span after the administration of *Aloe vera* compounds [37,71]. Of course, the antineoplastic efficiency of *Aloe vera* compounds is variable and depends on the concentration and type of compound tested, as well as the type and stage of tumor development [65,69]. In this context, the literature has shown that the efficacy of the compounds in ascending order is aloe-emodin, aloesin, octapeptide, and barbaloin. Thus, studies have shown the following trend in the inhibition of the growth and number of Ehrlich ascites carcinoma cells compared to the control group and thus the antineoplastic efficacy of these compounds, namely, aloesin < octapeptide < aloe-emodin < barbaloin [77].

**Figure 2 pharmaceutics-14-02004-f002:**
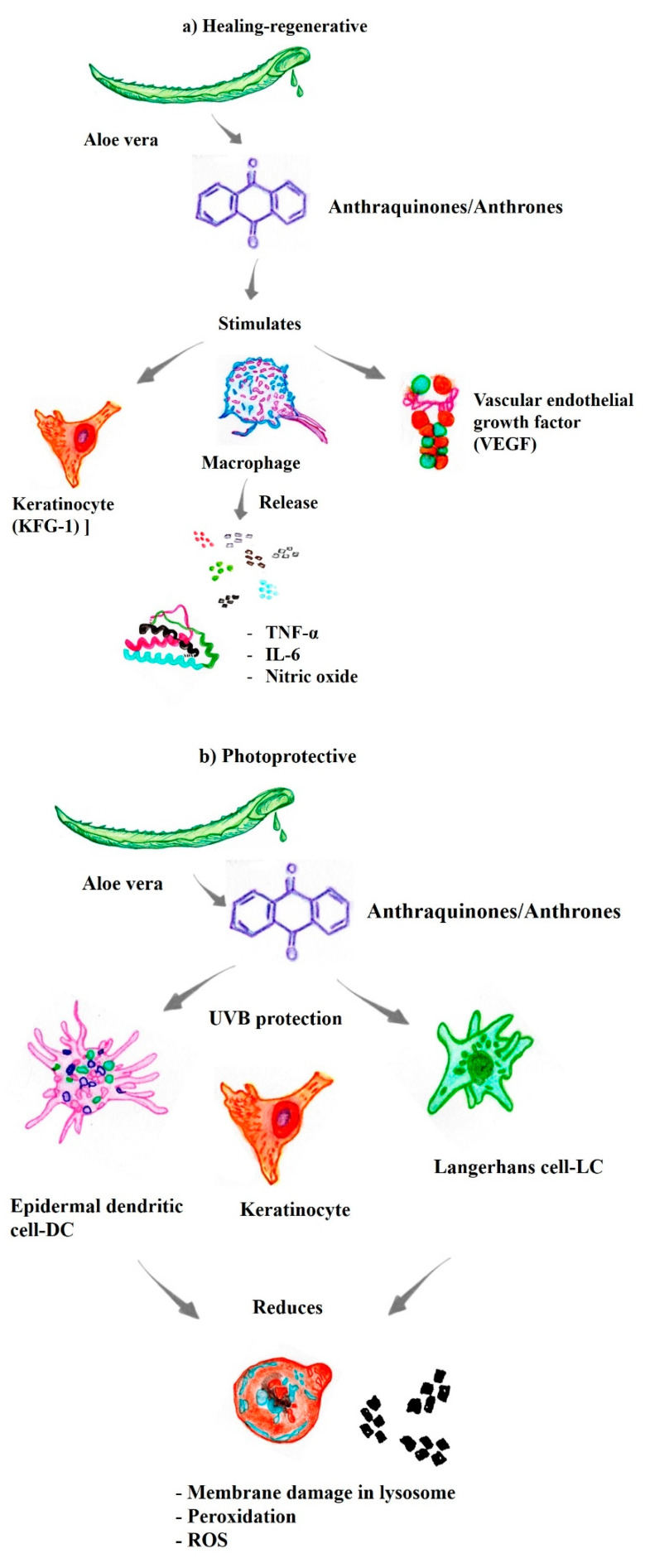
The substances present in aloe vera exhibit different biological properties of pharmacological/therapeutic interest. (**a**) Healing-regenerative [44,45,46]; (**b**) Photoprotective [37,51,52]; (**c**) Antiviral [55,56,57]; (**d**) DNA telomerase [59,61,62,63,64]; (**e**) Immunomodulatory [37,45,46,68,69,70]; (**f**) Anti-inflammatory and antiproliferative [45,46,47,48,59,63,68,72,74,75]; (**g**) Synergistic properties with antineoplastic drugs [42,74,78,79,80,81,82].

### 3.7. Synergistic Antitumor/Immunotherapeutic Activities 

*Aloe vera* acts synergistically with cisplatin to inhibit the proliferation of breast and cervical cancer cells. Research results have shown that the compound aloe-emodin enhances the activities of antineoplastic drugs such as tamoxifen, cisplatin, doxorubicin, cyclophosphamide, and 5-fluorouracil. In addition, some compounds, such as aloe-emodin, 7-hydroxy-2,5 dimethylchromone, and beta-sitosterol, have shown higher binding affinity for estrogen receptor alpha than the antineoplastic drug tamoxifen [16,42,67]. Aloe-emodin increases the radiosensitivity of cervical cancer cells, inhibiting their proliferation and, when combined with cisplatin, increasing antineoplastic activity in melanoma cells [69,74]. Furthermore, emodin sensitizes hepatocellular carcinoma cells to the antitumor drug sorafenib, a potent tyrosine kinase inhibitor (Figure 2g) [78]. 

Accordingly, aloe-emodin, when combined with phototherapies or photodynamic therapies, induces apoptosis in mucinoma cells, gastric cancer cells, and breast cancer cells [79,80]. Additionally, the compounds aloe-emodin, 7-hydroxy-2,5 dimethylchromone, and beta-sitosterol in *Aloe vera* act synergistically with cisplatin, inhibiting breast and cervical cancer cell proliferation and exhibiting a higher binding affinity for estrogen receptor alpha than standard tamoxifen [42,59]. 

Another study compared the administration of melatonin, a hormone related to the sleep-wake cycle, alone and in combination with *Aloe vera* in patients with advanced solid tumors, for which there is no effective standard of care. A positive response has been observed in patients treated with melatonin combined with *Aloe vera* compared to patients who received melatonin alone (2/24 versus 0/24). Furthermore, disease stability was achieved in patients treated with melatonin plus *Aloe vera* compared to groups receiving melatonin alone (12/24 versus 7/26). The percentage of patients with stable tumors was significantly higher in the groups treated with melatonin combined with *Aloe vera* than in the groups receiving melatonin alone (14/24 versus 7/26). In addition, the 1-year survival rate was significantly higher in patients treated with melatonin combined with *Aloe vera* (9/24 versus 4/26) [81].

Previous results have shown that acemannan combined with surgical and radiotherapy treatments produced a significant reduction in fibrosarcomas in canine and feline models. These animals presented recurrent failures in antitumor treatment, with a poor survival prognosis; however, after the combined treatment (radio/chemotherapy + acemannan), this trend was modified [73,82]. Similarly, the compound emodin has shown an unprecedented capacity to increase the antitumoral effects of gemcitabine on pancreatic cancer [65]. All these results indicate excellent prospects for reducing resistance to chemotherapy, as well as the various undesirable side effects that are common to antineoplastic treatment (Figure 2g).

## 4. Carbon-Based Nanomaterials with Antitumor Activities 

### 4.1. Carbon-Based Nanomaterials

All nanomaterials composed of carbon atoms are labeled carbon-based nanomaterials, and their classification depends on their geometric structure. Carbon nanostructures can have different shapes, such as planar, tubular, horn-shaped, spherical, and elliptical shapes [83,84]. Tubular NPs are termed carbon nanotubes; horn-shaped particles are known as nanohorns, and the spherical or ellipsoid NPs belong to the fullerene group [85]. Carbon nanomaterials, such as graphene, carbon nanotubes (CNTs), crystalline diamond, and diamond-like carbon, possess outstanding electrochemical properties, which have led to a broad set of applications. In this regard, carbon-based nanomaterials have attracted the attention of the academic and industrial sectors, mostly focusing on investigations associated with the health sciences [86]. 

In addition to the exceptional physical and chemical properties of these nanomaterials, such as excellent thermal and electrical conductivity, they can be chemically functionalized for specific applications, such as acting in molecular transport as the “molecular chassis” of therapeutic drugs. Thus, the use of nanomaterials is a fundamental method for the delivery of specific peptides and/or nucleic acids. In addition, these nanostructured agents are potential candidates for clinical diagnosis through bioimaging and/or biosensors [19,85,86].

### 4.2. Fullerenes

Since their discovery in 1985, fullerenes have attracted widespread attention from the investigation in basic research to applied sciences. They are hollow molecules composed entirely of carbon atoms in spherical, ellipsoidal, cylindrical, or tubular forms. Fullerenes are an allotropic modification of carbon, often termed as a molecular form of carbon, or carbon molecules. In addition, they may contain five or six-ring configurations and occasionally up to seven-ring configurations [87]. Fullerene NPs have received increasing attention due to their physical, electrical, chemical, and particularly antioxidant properties, which endow them with unique characteristics. They are known to be an efficient “free radical sponge” due to their strong electrophilicity, with the ability to accept up to six electrons [88]. 

For biological/biomedical applications, however, the hydrosolubility of pure fullerene, is very low, which poses a challenge. Several researchers have modified the fullerene surface to increase the hydrosolubility and biocompatibility. Functionalized or modified fullerenes present biocompatibility in addition to significant antitumor activity, which makes them excellent agents for applications in the field of cancer theranostics [89,90]. Therefore, an understanding of the antitumor mechanisms of fullerene NPs is very important for designing antitumor drugs with characteristics that confer low cytotoxicity and high specificity. 

Compared to antitumor drugs, fullerene-derived NPs are nontoxic, without multidrug resistance or cross-resistance to traditional anticancer chemistries. Moreover, fullerene NPs activate immunity, suppress angiogenesis, exert antioxidant effects, and reduce matrix metalloproteinase (MMP) production, which decreases blood vessel density and, consequently, the nutrient supply to the tumor microenvironment (Figure 3a) [90,91]. Fullerenes associated with paclitaxel (AF-1/paclitaxel), which is an antineoplastic drug, substantially reduce the proliferation of tumors derived from MCF-7 cells, with an inhibitory effect similar to that of paclitaxel, an NPs formulation used in clinical practice, with the commercial name Abraxane. Moreover, the pegylated stoichiometric conjugates DOX-C_60_ (1:1) and (2:1) exhibit antiproliferative activity against MCF-7 cancer cells, providing new insights into NPs used in nanomedicines for tumor treatment [89,92]. In addition, fullerenol synthesized from fullerene (C60) represents a promising candidate for the carrier effect as a “molecular chassis”, due to the enhanced cellular uptake and endosomal escape compared to other conventional polymeric vehicles. In this context, fullerenes possess unique structural features and properties suitable for cell–cell interactions, which are essential for biological and biomedical applications and depend on features such as size, charge, and functionalization, due to their efficient interactions with biomolecules [93].

Previous results have shown that the CM9-fullerene association exerts dose-dependent effects on the survival of leukemic-Jurkat cells, serving as an excellent agent in the treatment of acute leukemia [89]. Similarly, [Gd@C_82_(OH)_22_] nanoparticles show high antitumor efficiency in a variety of cancer xenograft models, such as liver cancer (H22 and HepG2 cells), human microvascular endothelial cells, and MCF-7 cells [89]. Interestingly, Gd@C_82_(OH)_22_ acts indirectly by “blocking” the tumor through adjustments of biomolecules that act together with the tumor microenvironment rather than by directly destroying the tumor [93,94]. Similarly, fullerene derivatives not only exert anticarcinogenic activity by regulating multiple steps in the crucial bioprocesses of carcinogenesis, including tumor growth and the metastasis of melanoma, breast carcinoma, prostate carcinoma, and leukemia, but also activate the patient’s immune system to eliminate these cancer cells (Figure 3a) [89,95]. 

### 4.3. Graphene and Graphene Oxide

Graphene is a two-dimensional allotropic form of carbon that is formed by a stack of single layers. Graphene has a hexagonal structure with sp^2^ hybridized carbon atoms and shows extraordinary characteristics, such as hardness, good thermal and electrical conductivity, and optical transmittance [96]. Graphene (G) and graphene oxide (GO) have been widely investigated for biomedical applications due to their exceptional qualities, such as a large surface-area-to-volume ratio, chemical/mechanical stability, excellent conductivity, and biocompatibility [97]. 

Graphene-based materials have shown efficiency in several medical areas, including imaging detection for cancer diagnosis, as a carrier effect in drug delivery by DDSs for immunotherapies, and tissue engineering for regenerative medicine [96,98]. Indeed, the large surface area-to-volume ratio of this nanomaterial, ease of functionalization, high capacity to transport therapeutic drugs/molecules, and the ability to induce ROS production have ranked these nanostructures in a unique position as agents next to antineoplastic immunotherapies (Figure 3b) [83,97].

Additionally, GO, the oxidized version of G, contains numerous oxygen-bearing functionalities on its surface, such as hydroxyl, carboxylic, and epoxide groups, which makes it more hydrophilic than G. Thus, in addition to presenting a large surface area-to-volume ratio, ease of surface modification and high capacity to transport therapeutic molecules, nanocomposites based on G and GO also have the potential to induce ROS production in tumor microenvironments [98,99]. These characteristics make these nanomaterials promising candidates for new DDS methodologies as early theranostic nanoplatforms to combat cancer cells [96,100]. Accordingly, these nanosystems have attracted the attention of different research groups and industries due to the aforementioned characteristics and improvements in tumor cell specificity, low adverse effects on healthy cells, and higher therapeutic uptake in the tumor microenvironment (Figure 3b) [99,101].

### 4.4. CNTs

CNTs are carbon allotropes with exceptional properties for technical applications. Currently, CNTs are widely used in the medical and biomedical fields as nanoplatforms for the transport of bioactive peptides, proteins, nucleic acids, and therapeutic drugs, aiming for specificity and therapeutic efficiency with decreased side effects [19,84,102]. CNTs are nanostructures with high stability, cytoprotective effects, and antioxidant activity. These characteristics have been widely explored due to additional characteristics, such as their excellent conductivity, which is very useful in the diagnosis of melanomas or skin infections [102,103]. 

Anticancer agents consisting of therapeutic biomolecules have been loaded into CNTs or onto the CNT surface through π−π stacking interactions between the therapeutic molecules and the pseudoaromatic double bonds of the graphene sheet. Another use is covalent functionalization or immobilization of the reactive functional groups present on the sidewalls of CNTs [104,105]. 

Hesabi et al. have investigated the use of anticancer drugs associated with CNTs, which showed good stability when combined with aminolevulinic acid [106]. In addition, studies have reported excellent results for the use of photothermal therapy with CNTs as a technique for cancer targeting, indicating that CNTs have potential as photothermal agents with the ability to obliterate malignant tumors upon exposure to NIR irradiation [107]. In murine models, phototherapy using CNTs induces complete tumor healing within 6 months, without any recurrence of the tumor mass [108,109]. These results have shown the efficiency of CNTs as a photothermal agent, with potential applications in future cancer therapy.

Sahoo et al. have functionalized MWCNTs and GO with the anticancer therapeutic drug camptothecin (CPT), and the resulting DDS was highly biocompatible and hydrophilic [110]. MWCNT-PVA and GO-PVA were encapsulated through π−π interactions and have shown the potential to kill cancer cells in both melanoma and breast cancer. Additionally, cisplatin, another therapeutic molecule, can be loaded onto SWCNTs with a diameter of 1.3 to 1.6 nm. Doxorubicin has been loaded onto CNTs through π−π stacking interactions by mixing the drug with an aqueous CNT dispersion stabilized by a nonionic surfactant, e.g., Pluronic F127 [111,112]. These results indicate that MWCNTs can enhances the delivery of therapeutic drugs, including antitumor drugs such as doxorubicin, by improving the cellular uptake of these therapeutic compounds without causing cytotoxicity toward healthy cells (Figure 3c) [112].

In addition, carbon nanomaterials have attracted attention in the areas of diagnostics and biomarkers, especially in the detection of phosphorylated proteins and peptides in biological samples. The characterization of phospholipids and phosphorylated proteins provides important information on physiological and pathological states associated with the removal or addition of phosphate groups, which control cellular functions [113,114]. However, phosphorylated biomolecules are labile, which poses a challenge in proteomic analyses. In this context, carbon nanomaterials have been used for protein immobilization via ion affinity chromatography (IMAC) and metal oxide affinity chromatography (MOAC) with excellent detection of the compounds of interest, avoiding particle aggregation due to contact affinity and with high sample specificity. Additionally, carbon nanomaterials with low porosity are advantageous because they do not require pore entry and exit for analytes to bind, a characteristic that classifies them as excellent candidates for biomarker detection, especially due to their efficiency, selectivity, and analytical reproducibility [114,115].

However, one of the greatest obstacles of CNT applications in biological/biomedical applications is the possible incompatibility, limited biodistribution, and toxicity of these nanomaterials. In this context, the understanding of the biological interactions that occur between CNTs and biomolecules, as well as the processes of protein adsorption on the surfaces of these nanomaterials and an understanding of interactions of these proteins, for example, the formation of corona on the surface of the CNT, are essential factors when considering the possibility of modulating these integrative kinetic behaviors of these nanoparticles, affecting their kinetics, uptake, biodistribution, and, subsequently, the immune response to these nanomaterials [84,105,116]. Thus, the understanding of all these factors together is essential in the search for standards to establish best practices for the design and functionalization of these carbon nanomaterials for their direct implementation in vivo.

Another factor that has been discussed by committees related to the best production of CNTs is the search for a common sense guideline for the production standards and classification of these nanomaterials for direct in vivo applications [117].

### 4.5. Carbon Nanofibers and Nanohorns

Carbon nanofibers (CNFs) are carbon filaments with sizes ranging from 5 to 100 μm in length and 5 to 100 nm in diameter. The filament core of CNFs may have a hollow and continuous morphological configuration, giving rise to tubes (or nanotubes); alternatively, it maby be hollow and discontinuous, with a structure resembling bamboo, or completely filled to form a rod-like shape. Thus, the solid carbon fibers of CNFs differ from CNTs due to the absence of a hollow cavity [118,119]. 

CNFs are linear, noncontinuous sp^2^-based filaments, unlike continuous, multimicrometer diameter carbon fibers. CNFs are characterized by flexibility, a relatively large specific surface area, and a uniform magnitude and distribution. The large surface area and porosity of CNFs are favorable characteristics for analytical applications, especially biomedical applications. With a specific surface area of approximately 1877 m^2^ g^−1^, CNFs have one of the highest specific areas ever reported for nanostructured materials [120]. The specific area of CNFs may be significantly modified for specific applications by removing the most reactive carbon atoms from the structure, making them promising nanomaterials, especially for biological and biomedical applications, including biosensors, tissue engineering, and carrier-DDSs (Figure 3d) [121]. 

The unique properties of CNFs, combined with active sites such as the exposed graphite edge regions, are available for chemical and physical interactions with high surface areas (300–700 m^2^g^−1^), which represents the entire surface area of their chemically active surface. Thus, the surface area of this nanomaterial is another factor that should be considered for DDS applications [120,121]. 

In addition, carbon nanohorns (CNHs), or single-walled carbon nanohorns (SWNHs), are composed of a single graphene sheet that ranges between 2 and 5 nm in diameter and 40 and 50 nm in length. CNHs represent a structure analogous to CNTs except that they are closed at one end, forming a cone-shaped or “horn” cap. They tend to form spherical aggregates larger than 100 nm. Carbon nanohorms have been explored as a potential anticancer agent for the delivery of antineoplastic drugs known from clinical practice, including cisplatin and dexamethasone [122]. 

CNHs have also been explored as anti-inflammatory agents, for example, by administration combined with prednisolone (PLS), a glucocorticoid anti-inflammatory agent. Thus, PSL showed improved adsorption and hydrophilicity when combined with oxidized (oxSWNHs) and single-walled CNHs [122,123]. In this context, PSL was adsorbed on both the outside and inside of oxSWNHs and released rapidly within a few hours and then slowly in approximately 1 day from the respective sites. Additionally, Wang et al. relabeled a multifunctional and efficient DDS of oxSWNHs, which showed improvements in antitumor effects when combined with methotrexate [116,118]. Accordingly, among all the different carbon structures employed in carrier-DDS systems, single-walled nanohorns are the most suitable due to their small size, ease of synthesis, and low cytotoxicity after functionalization (Figure 3e) [121,122].

### 4.6. Carbon Quantum Dots

Carbon quantum dots (CQDs) have received increasing attention in recent years due to their excellent properties for replacing conventional QDs in various applications. CQDs are new zero-dimensional carbon-based NPs with mostly sp^2^ hybridization, followed by sp^3^ hybridization [124]. CQDs exhibit exceptional optical and fluorescent characteristics, high aqueous solubility, easy and inexpensive synthesis, simple surface functionalization, and high optical and thermal photostability, which are essential for their use in various applications, especially in medical/biological areas [125]. 

Other properties of CQDs, such as superior aqueous solubility, chemical stability, low cytotoxicity, and high emission quantum yield, make them excellent nanomaterials for biomedical applications, particularly bioimaging [126]. Furthermore, studies have shown that CQDs exert antiproliferative, cytotoxic, and apoptotic effects on cancer cells, suggesting a potential avenue for antineoplastic treatment. CQDs have been shown to possess antiproliferative and antitumor activity against MCF-7 breast cancer cells, thus representing a promising agent for cancer immunotherapies (Figure 3f) [127]. CQDs interact with cancer cells to form ROS, resulting in the death of these cells. Furthermore, CQDs are promising in medical imaging/diagnostics, as they are internalized by cancer cells and visualized using fluorescence microscopy [128,129]. However, the ability of CQDs to exert anticancer effects has been attributed to different molecular mechanisms, which are not yet fully understood.

### 4.7. Nanodiamonds

Nanodiamonds (NDs) consist of an sp^3^-hybridized carbon nanonetwork with exceptional stability. The difference between NDs and other carbon nanomaterials is the mode of grouping/bonding between these carbon atoms. The surfaces of NDs contain various functional groups, a key feature for coupling with therapeutic molecules and chemically modifying the nanomaterial surface [130,131]. 

**Figure 3 pharmaceutics-14-02004-f003:**
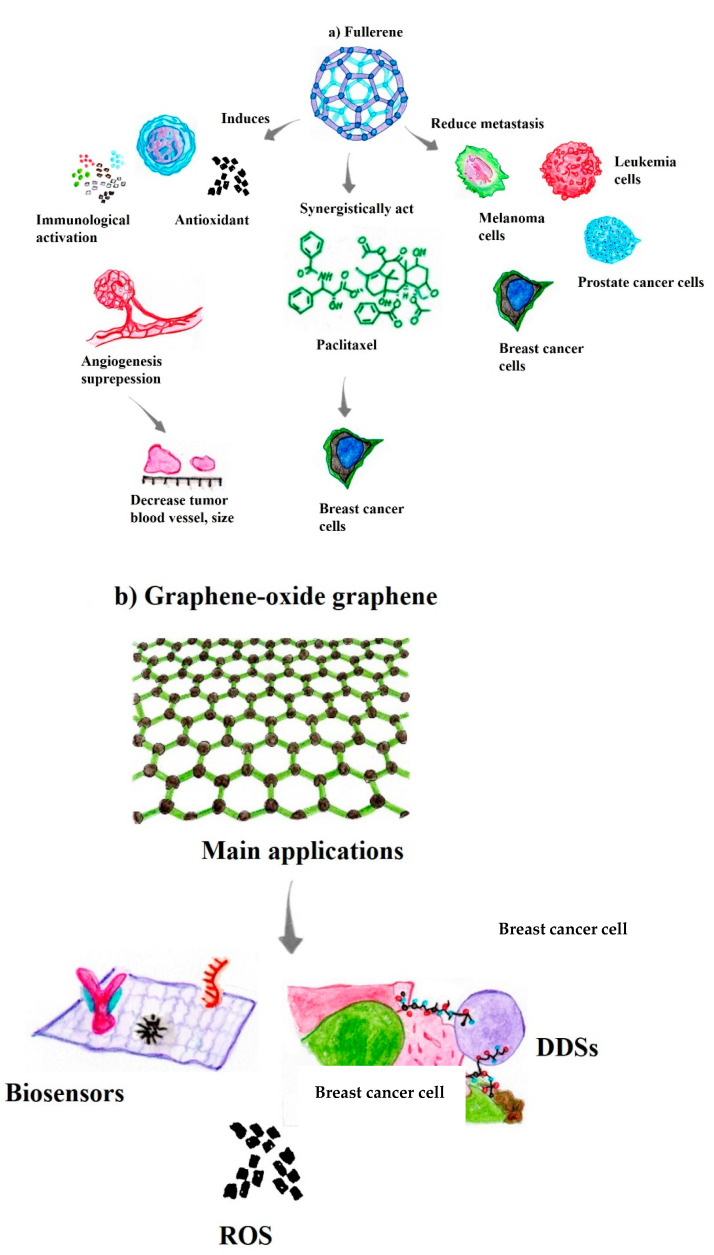
Carbon-based nanomaterials and their main biological activities for therapeutic and theranostic applications. (**a**) Fullerene [89,90,91,92,93,94,95]; (**b**) Graphene, graphene oxide [96,98,100]; (**c**) Carbon nanotubes [104,105,106,108,110,111]; (**d**) Carbon nanofibers [114,116,117,118]; (**e**) Carbon nanohorm [120,121,122,123]; (**f**) Quantum dots [125,126,128,131]; (**g**) Nanodiamonds [128,129,130,131].

Functional groups on the surface of NDs promote a diverse range of desirable properties, such as excellent thermal conductivity and physical and mechanical properties, making these nanomaterials highly interesting for various applications. NDs have several superior properties related to their optical and chemical stability, excellent physical properties, and the ability to form conjugated bonds with drugs for medical applications. Accordingly, NDs are excellent agents for the molecular transport of antineoplastic therapeutic drugs as DDS [132]. 

The main advantages of NDs in medical applications are related to their high compatibility with noncancerous cells. In addition, NDs present great potential for surface functionalization with antineoplastic drugs, with a relatively low synthesis cost. Another factor to consider is that NDs can significantly optimize the treatment efficacy of the transported drugs, allowing anticancer drugs to exhibit higher efficacy even at lower doses and consequently decreasing the side effects of the therapeutic drugs [133,134]. 

Functionalized NDs transport therapeutic molecules that accumulate in the tumor microenvironment, which entails impaired therapeutic drug efflux from cancer cells. In addition, NDs functionalized with or coupled to antineoplastic molecules, such as doxorubicin, increase the solubility and therapeutic efficacy and reduce the organ and systemic cytotoxicity of the drugs [129]. NDs increase nucleic acid delivery up to 70-fold in cancer treatment, improving the efficacy of the lipid-based molecule lipofectamine. In addition, the functionalization of NDs for the carrier-DDS effect induces better penetration of anticancer drugs, including hydroxyurea, into the tumor microenvironment. Moreover, functionalized NDs have been shown to increase the solubility of paclitaxel and decrease the toxicity of cisplatin in noncancer cells while simultaneously increasing the toxicity toward cancer cells (Figure 3g) [134,135]. In this context, NDs plays a key role in the design of new nanoformulations in combination with antineoplastic molecules, representing encouraging prospects for new antitumor therapeutics.

## 5. Advances and Challenges in the Use of *Aloe vera* Combined with Carbon Nanomaterials in Antitumor Applications 

The synthesis of NPs involves physical methods that are time-consuming, costly, and energy-intensive and chemical methods that are fast but involve the use of toxic chemicals, resulting in harmful byproducts that negatively affect the ecosystem. Thus, these limitations in the development of NPs have shifted researchers’ attention to biogenic synthesis or “green” nanotechnology, which involves production using plant materials including roots, shoots, and leaves, with lower costs and greater biocompatibility. Studies show that the synthesis of “green” NPs composed of tellurium (TE) associated with *Aloe vera* (AV-TeNPs), induced a significant reduction in the growth of melanoma cells without causing cytotoxicity to human fibroblasts [136]. Green nanotechnology contains numerous functional groups derived from phytochemical constituents found, for example, in *Aloe vera*, including polysaccharides, proteins, enzymes, and vitamins, among other phenolic compounds, which can kill tumor cells at concentrations lower than usual, i.e., 25 µg/mL [136,137]. In this sense, several studies have been performed with different plant species, especially the compounds present in *Aloe vera*, which have long been known for their antibacterial, anti-inflammatory, healing, and antineoplastic properties, among other numerous pharmacological and therapeutic properties. Moreover, these phytoconstituents act synergistically with other chemical compounds already used in antitumor therapy, potentiating the specific and desired response [78,138].

In addition, aloe-emodin (AE, 1,8-dihydroxy-3-hydroxymethylanthraquinone), an important bioactive anthraquinone compound present in *Aloe vera*, exerts anti-proliferative and apoptotic effects on different tumor cell lines, with various pharmacological applications [42,139]. However, the translational use of this compound is limited due to issues such as rapid degradability and low bioavailability. Thus, Li et al. prepared AE-based nanoparticles associated with lactic-co-glycolic acid (nanoAE) that induced significantly reduced cancer cell growth and proliferation, and activated apoptosis, as evidenced by cleavage of Caspases-3, 8, and 9, increased ROS production, and mitogen-activated protein kinases (MAPKs), P13K/AKT, and mTOR inactivation. The nanoformulation (nanoAE) inhibited cell growth and proliferation at G1/S and G2/M phases by altering the expression levels of regulatory genes such as p53, p21, and cyclins, which are important DNA damage validation markers [65,137]. 

Moreover, the formulation exerted inhibitory effects on squamous cell lung carcinoma tumor cells but showed no cytotoxicity toward healthy cells, indicating the efficiency and specificity of this nanoformulation. Thus, advances in the use of AE include not only applications in different tumor cells to induce apoptosis, interrupt the cell cycle and metastasis, and improve immune activity but also as a sensitizing agent in antitumor chemotherapies and radiotherapies, which may represent a new and promising strategy for antineoplastic therapeutic management of different tumor types [54,65,68]. 

In addition, several in vitro and in vivo studies have reported promising antitumor effects on different tumor types related to the antioxidant and antineoplastic action of *Aloe vera* phytotherapeutic compounds. The properties inherent in *Aloe vera* suggest its enormous biological and biomedical potential as an immunotherapeutic that might be explored using different approaches in medical and biological research [37]. Some compounds in *Aloe vera* exhibit specificity regarding antioxidant and antitumor activity against different cancer types [38,47,48]. 

Studies investigating the specificity of these compounds and their relationships with tumor development and/or inhibition are essential for directing both research design and funding. In this context, the immunotherapeutic prospects for the use of *Aloe vera* are encouraging, since the plant has distinct properties that can be exploited by the medical and pharmaceutical industries, especially compounds with antioxidant, healing, antibacterial, antimicrobial, and antineoplastic activities [37,52,54]. 

*Aloe vera* is an excellent candidate for antineoplastic therapy because of its wide range of positive biological effects/activities, the low investment related to its production in different countries, especially those with limited financial resources, and the good prospects for its distribution and ease of use, which are fundamental factors for the accessibility of antitumor treatment [11]. Thus, studies are necessary to improve the existing immunotherapies and medicaments, aiming at the development of new drugs with increased efficacy and antitumor specificity and easy access to the population. Several in vitro and in vivo studies have characterized and measured the antioxidant and antineoplastic properties of *Aloe vera* in different tumor types [38,48,68].

In vivo studies in murine models have documented the antitumor and antioxidant properties of *Aloe vera* against Ehrlich carcinoma and ascites in distinct cancer lineages [140]. Other studies have shown positive results from the use of *Aloe vera* compounds, especially those derived from anthraquinones, in different cancer lineages, including lung, duodenum, prostate, breast, and colon cancers and melanomas. Similar results have been obtained in the treatment of melanomas using AE [59,61,141]. In this context, *Aloe vera* compounds represent a promising option for combined and/or alternative immunotherapies, with great potential for the treatment of tumors as innovative approaches to traditional immunotherapies not only for melanomas but also for different tumor lineages. 

In addition, the use of nanomaterials associated with therapeutic/antineoplastic molecules, including C_60_-paclitaxel and fullerenol conjugated with cisplatin (CDDP), suppresses tumor growth without cytotoxicity to healthy tissues. C_60_(OH)_n_/C_70_(OH)_n_ (n = 18–20) inhibits the growth of both Hep-2 laryngeal and cervical cancer [87,90]. 

Some challenges related to the categorization and classification of *Aloe vera* compounds have not been fully elucidated in terms of possible cytotoxic mechanisms against healthy cells, which might be mediated by the action of anthraquinones. However, *Aloe vera* does not exert carcinogenic or cytotoxic effects on embryonic or reproductive tissues [59,73]. Another issue is that AE, due to the presence of hydroxyl groups in its structure, has the potential to trigger some tumor-promoting activity. In vivo and in vitro studies conducted with substances possessing the same chemical nature have shown that many of these tumorigenic activities involve different biochemical pathways of the organism, such as DNA replication, modulation of oncogenes, and accumulation of mutations in the genetic code [48,140,141]. 

Thus, the use of plants for therapeutic purposes should be applied with caution since side effects may occur due to the presence of anthraquinones as an example. In addition, anthraquinones stimulate the gastrointestinal system, which may cause reflexes in the uterine musculature, inducing abortions in pregnant women. Other problems associated with the excessive use of *Aloe vera* include cramps, nausea, diarrhea, damage to the neuromuscular system, and chronic kidney damage [37,142,143]. 

In this context, the main problems associated with the use of *Aloe vera* in antineoplastic treatment are related to the identification and characterization of some as yet unknown compounds present in its extract or gel. Another issue to be considered is that herbal medicines based on *Aloe vera* are volatile and degrade easily. In this context, *Aloe vera* in natura has higher biological activity than the same compounds refrigerated and/or stored for a long period [41,141,142]. Thus, new studies aiming to understand how these compounds behave and why changes occur in their biological or biochemical activity over time are fundamental for antineoplastic applications. Accordingly, studying ways to preserve these properties longer for possible drug development is an area of great pharmacological and commercial interest. 

Another relevant concern is that most studies with *Aloe vera* have been performed with limited data, and these studies must be expanded to larger populations to obtain more accurate data that will confirm the therapeutic properties reported in studies with small populations [75,143]. Additionally, the tests should be conducted specifically for each type of cancer, since the biological foundations of cancer differ, despite following the same processes of tumor formation, because of the tissues from which they are derived and the time at which they begin to develop [3,67]. 

Thus, evaluations of the specificity, efficiency, and biosafety of new therapeutic molecules based on carbon nanomaterials and combined with phytotherapeutic compounds are essential and urgently needed. Accordingly, the construction of antineoplastic nanoparticles combined with *Aloe vera* compounds to restrict their cytotoxic activities to cancerous cells is desirable [144,145]. The use of carbon nanomaterials in combination with *Aloe vera* compounds has great immunotherapeutic potential to address these challenges associated with antineoplastic treatments. 

The search for nanotechnological molecules and efficient chemical and physical modification methods to overcome these limitations is fundamental to improving the efficacy of *Aloe vera* as an antineoplastic agent. Thus, incorporating phytotherapeutic molecules into DDSs represents a unique opportunity to improve the uptake and efficacy of the antineoplastic drug as well as patient compliance, thereby improving treatment responses [38,48,59].

## 6. Discussion and Perspectives on the Use of *Aloe vera* Combined with Carbon Nanomaterials in Antitumor Immunotherapies

Melanoma is the deadliest type of cancer, and studies aiming to improve its therapeutic options are urgently needed [3,4]. The use of *Aloe vera* as an antitumor agent, especially the compound AE, presents an antitumor activity that begins with the inhibition of unrestrained cell proliferation and culminates in the regulation of the immune response of the organism. The compound emodin acts directly on several metabolic pathways, including apoptosis and the production of fundamental cellular mediators of the immune response of the organism, such as TNF-α, FASL, GM-CSF, NF-KB, and ROS. In addition, emodin acts on the cell cycle and transcription factors by modulating the expression of genes such as BRAF, which are related to tumor development [47,68,146]. 

Additionally, AE exerts selective cytotoxic effects on certain tumor types that are related to specific cellular uptake, such as neuroectodermal tumors [66]. Accordingly, a wide range of applications of AE have been described in the treatment and/or prevention of melanomas, particularly in methodologies using carbon-based nanomaterials [47,136,147]. The synergistic effects of the interactions of the compounds present in *Aloe vera* with bioactive substances combined with nanomaterials, such as cisplatin, result in excellent antineoplastic activities through the immunomodulation of tumors [78,148].

In this context, the recognition and understanding of these interactions of *Aloe vera* compounds and nanomaterials are of utmost importance for the efficient design of these nanoformulations. The combination of nanomaterials with antitumor compounds promotes not only the exploration of synergistic effects but also improvements in therapeutic drug delivery by DDSs, as well as theranostics against tumors of different origins [43,136,148]. The advantages of using these DDS models include enhanced biological activity of the drug and the possibility of combining multiple drugs of interest, i.e., the development of theranostic nanoplatforms, protection against biochemical degradation, and specific and continuous drug delivery within the tumor microenvironment [96,149,150].

Studies have shown improved performance associated with the application of NPs delivery mechanisms, and some of these drugs have been approved by the FDA and are already part of antineoplastic clinical practice, including the drugs doxorubicin, cisplatin, paclitaxel, vincristine, and cyclodextrin. The most commonly used nanomaterials in these nanoformulations are liposomes, dendrimers, CNTs, inorganic NPs, micelles, magnetic NPs, and CQDs [18,92,93,112,151]. 

*Aloe vera* compounds associated with poly (lactic-co-glycolic acid)-based nanoparticles (nanoAE) produce excellent antitumor responses, with a significant repression of cancer cell proliferation and direct inductikon of cell cycle arrest and apoptosis of these cells. These results highlight the cleavage mechanisms of Caspase-3, poly (ADP-ribose) polymerase (PARP), Caspase-8, and Caspase-9 [152]. Interestingly, this uptake of nanoformulations, in addition to being favored due to the small size of the structure, still reduces unwanted side effects, enabling the targeted design of these nanoplatforms that facilitates full recognition and specificity. Accordingly, nanoformulations with *Aloe vera* effectively arrest the cell cycle in the replicative G1/S and G2/M phases, increasing the apoptosis of these cancer cells [12,68,153]. Cell proliferation is governed by a complete cell cycle, which includes regulation by cyclin-dependent kinases (CDKs) and CDK inhibitor proteins. In addition, p53 is a tumor suppressor gene that regulates the expression of genes involved in the cell cycle, and p21 prevents DNA synthesis, arresting cell growth [154]. 

Studies have shown that the use of nanoformulations with *Aloe vera* compounds (nanoAE) substantially reduces the expression levels of p53, p21, and Cyclin B/E, which are important checkpoint markers of cell damage during the DNA replicative cycle. In this regard, *Aloe vera*-based nanoformulations activate different metabolic pathways linked to apoptosis, including but not limited to Bax, Bad, and Caspase signaling. Thus, nanoformulations containing compounds from *Aloe vera* induce ROS production, followed by increased levels of p-ERK1/2, p-p38, and p-JNK [153,155].

Additionally, mechanisms that regulate phosphatidylinositol-3 kinase (PI3K)/AKT are dysregulated in various types of cancer, and ROS are related to PI3K/AKT signaling [156]. In vivo studies have shown that PI3K/AKT is inactivated by nanoAE at higher levels than free AE. Thus, the AE/nanoAE nanoformulation exhibited greater tumor-suppressive capacity, without exhibiting cytotoxicity to liver or kidney cells [152]. In this context, the synergistic effects of nanoparticles combined with *Aloe vera* have the advantage of increased specificity in suppressing the growth of tumor cells through the activation of apoptotic mechanisms, such as MAPKs-caspases, and ROS generation-dependent pathways, which may represent an excellent therapeutic strategy for cancer management and new antineoplastic targeting [138,148].

In fact, phytochemicals are the main source for the development of innovative drugs that can be used in both cancer prevention and treatment. Thus, in vitro and preclinical studies have shown that NPs are effective in delivering chemotherapeutics and natural compounds that provide curative and sensitizing actions. Medicinal plants are highly sought after due to their inhibitory effect on cancer cells. Although the development of nanopharmaceuticals is still in its infancy compared with established drugs, the development of natural bioactive chemicals using nanotechnology has been the subject of several studies with herbal medicines aiming to improve their efficacy and mitigate their side effects [157].

One of the limitations of herbal medicines is that they may have multiple targets, affecting not only target cells, but also a wide range of normal cells. Another issue is that when herbal medicine is used with other drugs, it may decrease the bioavailability of chemotherapeutic drugs. Therefore, pharmacological interactions between herbal drugs and other FDA-approved traditional drugs should be thoroughly explored to identify the most effective additive mixtures. In addition, a limiting factor is the incipient database of the specific interactions of these drugs with other prescription drugs where several compounds show structural similarity but confirmation of their structure–function relationship is not well defined [157]. Therefore, the interaction of anticancer drugs with herbal medicines must be studied for the proper development and use of herbal medicines associated with nanomaterials. Although traditional medicine has established surveillance systems, herbal-ayurvedic treatments lack adequate systems for monitoring, data analysis, objectivity, measurement tools, and therapeutic standardization. In this regard, pharmaceutical regulations for safety, quality, and efficacy must be developed and managed [157]. 

Notably, all these challenges, including toxicity, nonspecificity, biodistribution, biomolecular interactions, and pharmacokinetics, might be mitigated through the possibility of incorporating carbon nanomaterials with phytochemicals. Functionalization with different organic groups, polymers, or alternative structures prevents aggregation and reduces toxicity in human cells. Therefore, the use of *Aloe vera* (AV), which has been used since ancient times as a treatment for skin cuts, burns, skin diseases and worm infections, has been used as an anticancer treatment. Some progress has been achieved, demonstrating that the presence of AVTeNPs disrupts melanoma cell membranes; which exhibit morphologies similar to cell apoptotic mechanisms and are proposed to be attributed to excess ROS levels [136].

## 7. Future Directions and Conclusions 

Recent advances suggest that nanotechnology-driven revolutions exploring the use of carbon-based NPs may be effective in combating tumorigenesis and angiogenesis, serving as a promising alternative to conventional antineoplastic treatments. Thus, among carbon-based NPs, including carbon dots, graphene, nanodiamonds, and carbon nanofibers, CNTs are at the forefront of research due to their immunomodulatory, antimicrobial, and antineoplastic properties. 

In their elemental form, CNTs are immunomodulatory, with potential for immune activation of specific signaling pathways and effective adjuvant action. Additionally, the functionalization and/or association of these nanomaterials with other compounds and biomolecules increases their immunomodulatory properties and, consequently, antineoplastic and angiogenic properties. These immunomodulatory properties of carbon-based nanomaterials are mainly due to the high surface area-volume ratio of these NPs, which allows greater interaction and accessibility to the tumor microenvironment.

Thus, several carbon-based nanomaterials and updated versions are constantly being developed, particularly nanomaterials functionalized with biomolecules, with proven efficacy for theranostic use in different types of tumors, including melanoma. Skin cancer is a fast-growing type of cancer that requires rapid and accurate diagnostic tools. However, despite the exceptional role of conventional methods in reducing the mortality rates of patients with melanoma, the limitations of these methods for clinical use and recent advances in optical and electrochemical nanosensors have led to innovative approaches to melanoma therapeutics in diagnosis and immunotherapy. Thus, nanomaterials functionalized with therapeutic molecules might better meet this demand. Accordingly, the antineoplastic activity of *Aloe vera*, especially the compounds aloin, AE, and acemannan, exerts positive effects on antineoplastic control in different types of cancer because of their antioxidant activities, with the ability to reduce the growth of different tumor types, particularly melanoma. 

As *Aloe vera* compounds affect several signaling pathways to exert antitumor activities, their combination with carbon-based nanomaterials that act concomitantly with existing antineoplastics represents an exceptional pathway for the development of new nanoformulations with increased capacity antineoplastic specificity. New studies need guidance for the development of innovative nanoformulations that act synergistically in antitumor therapy. Thus, the use of compounds derived from *Aloe vera* combined with carbon NPs represents the Holy Grail of an excellent antineoplastic alternative that can be combined with existing therapies for the treatment of different types of tumors, including melanoma. 

Our work provides insight into the most relevant aspects of the research of carbon-based nanomaterials and the possibilities, advances, and challenges associated with the use of phytotherapeutic compounds from *Aloe vera* in antineoplastic therapy by discussing and presenting important results that may be useful in new scientific designs for the management of antineoplastic therapy of different tumors, including melanoma.

Thus, standardization of the analysis of the antitumor effects of compounds from *Aloe vera* and the quantification of comparative parameters when combined with nanomaterials are necessary and urgently needed. Despite the numerous encouraging results obtained for several types of cancer, functionalization processes for these nanoparticles must be established that not only allow greater efficiency in the design and development of immunotherapies based on compounds from *Aloe vera* but also act synergistically with other therapeutic drugs already in clinical use to optimize the antitumor response. 

In conclusion, establishing a better understanding of these synergistic processes of carbon-based nanoformulations combined with herbal compounds, such as combinations of *Aloe vera* and antineoplastic drugs, represents an unprecedented opportunity for antineoplastic pharmacotherapeutics and theranostics that will provide hope in the fight against different types of cancer, including melanoma.

## Figures and Tables

**Figure 1 pharmaceutics-14-02004-f001:**
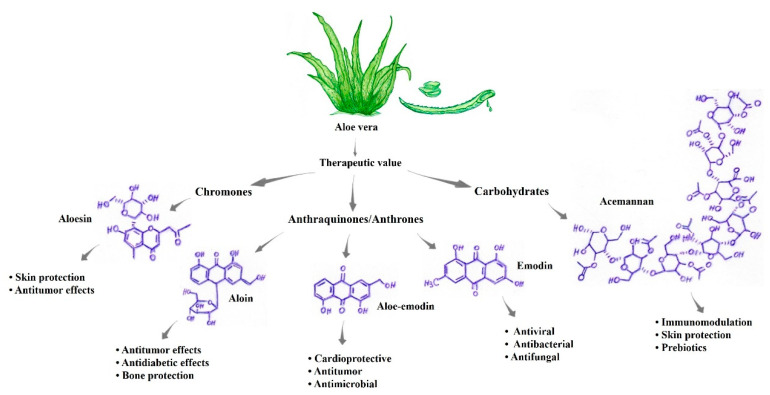
Active components present in aloe vera with high and recognized therapeutic value [37,38,40,41].

**Table 1 pharmaceutics-14-02004-t001:** Examples of genes related to the melanoma tumorigenesis process.

Gene and Chromosome	Mutations	Product	Function [Reference]
BRAF 7q34	T1799A, V600E, K601E, G469A	Serine/threonine-protein kinase	Activates a RAF/MEK/ERK signal transduction cascade, responsible for the cell division process [28]
NRAS 1p13	Q61R _Q61K	Guanosine triphosphate/diphosphate binding proteins	Extracellular signal transducer, responsible for communication between the cell membrane and nucleus [29]
NF1 17q11	Several polymorphisms	Cytoplasmic protein (Neurofibromin a)	Tumor suppressor [30]
TERT 5p1	rs2853669	Catalytic subunit of telomerase (ribonucleoprotein complex)	Maintenance of telomere length [28]
CDKN2A 9p21 9p13-p22	Several polymorphisms	p16INK4A and p14ARF proteins	Transcription of tumor suppressor proteins (p16INK4A and p14ARF) [31]
TP53 17p13	Several polymorphisms	Nuclear phosphoprotein	Factor controlling the repair of damaged DNA and cell growth control by regulating transcription [32]
CDK4 12q14	Several polymorphisms	Kinases forming heterodimers with D-type cyclins	Regulates the transition between G1 and S phases in the cell replication cycle [33]
AKT 14q32	Several polymorphisms	Protein kinase B alpha, beta, and gamma	Regulates various cellular functions, such as cell proliferation, survival, metabolism, and angiogenesis in normal and malignant cells [32]

## Data Availability

Not applicable.

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
