# Peer review of "Effects of Carbon Nanomaterials and Aloe vera on Melanomas—Where Are We? Recent Updates"

_pharmaceutics, 2022, doi:10.3390/pharmaceutics14102004_

Round 1

Reviewer 1 Report

This is an interesting and thorough review that will be of broad interest, thus I recommend its publication. I enjooyed reading it and found the illkustrations very attractive and distinctive. I found only a few minor, but important, corrections:

1. line 58, page 2, the diameter for melanomas should be in "mm" and not "nm"

2. Section 4.2. All chemical formulas MUST have numbers in subscript, such as C60 and Gd@C82(OH)22

3. Likewise please correct CH2Cl2 line 448

3. the CNF drawing (3d) is misleading and looks like a bamboo-type MWCNT, rather than a CNF. A CNF could be drawn as parallel CNTs, like a bundle of spaghettis, see for instance the TEM image here: https://www.materialstoday.com/carbon/news/highquality-fibers-from-carbon-nanotubes/

4. in the nanocarbon part, there should be  a mention about some general concerns, such as pertaining the biocorona and consequent effects on pharmacokinetics and immune response, the lack of unified standards, and the concerns for toxicity. For inspiration, the authors could have a look at the last paragraph of the introduction of this work: https://www.mdpi.com/2227-9059/9/5/570/htm

5. another aspect that is relevant for this review is the fact that nanocarbons are attracting interest also as enrichment phases for biomarkers detection, such as labile phosphopeptides for early diagnosis of cancer. A couple of sentences about this should be included. Traditionally, porous carbon has been used for this end, especially in composites with metals and metal oxides for affinity chromatography. Recent works are finding better performance with various nanocarbons:

https://link.springer.com/article/10.1007/s12274-020-2620-4

https://www.sciencedirect.com/science/article/pii/S1570023221003329

https://chemistry-europe.onlinelibrary.wiley.com/doi/full/10.1002/slct.202004650

A general review on the concept:

https://www.mdpi.com/2079-4991/12/11/1845

Author Response

Reviewer #1

The reviewer mentions that:

- This is an interesting and thorough review that will be of broad interest, thus I recommend its publication. I enjooyed reading it and found the illkustrations very attractive and distinctive. I found only a few minor, but important, corrections:

  • line 58, page 2, the diameter for melanomas should be in "mm" and not "nm"

R: The requested change has been completed, as noted on page 2, line 587.

  • Section 4.2. All chemical formulas MUST have numbers in subscript, such as C60 and Gd@C82(OH)22

R: As requested, the numbers in all chemical formulas are now shown in subscript, both throughout the text and in the tables where they are found (page 15, lines 477 and 479 and Table 2, pages 7, lines 202).

  • Likewise please correct CH2Cl2line 448

R: As requested, this chemical formula was also corrected (page 12, line 365)

  • the CNF drawing (3d) is misleading and looks like a bamboo-type MWCNT, rather than a CNF. A CNF could be drawn as parallel CNTs, like a bundle of spaghettis, see for instance the TEM image here: https://www.materialstoday.com/carbon/news/highquality-fibers-from-carbon-nanotubes/

R: As requested, we generated new drawings of the CNTs (3D) in parallel, similar to a spaghetti bundle based on the recommended references now, the drawing has been revised based on the TEM image.

  • /6 in the nanocarbon part, there should be  a mention about some general concerns, such as pertaining the biocorona and consequent effects on pharmacokinetics and immune response, the lack of unified standards, and the concerns for toxicity. For inspiration, the authors could have a look at the last paragraph of the introduction of this work: https://www.mdpi.com/2227-9059/9/5/570/htm
  • another aspect that is relevant for this review is the fact that nanocarbons are attracting interest also as enrichment phases for biomarkers detection, such as labile phosphopeptides for early diagnosis of cancer. A couple of sentences about this should be included. Traditionally, porous carbon has been used for this end, especially in composites with metals and metal oxides for affinity chromatography. Recent works are finding better performance with various nanocarbons:
  • https://link.springer.com/article/10.1007/s12274-020-2620-4
  • https://www.sciencedirect.com/science/article/pii/S1570023221003329
  • https://chemistry-europe.onlinelibrary.wiley.com/doi/full/10.1002/slct.202004650
  • A general review on the concept:
  • https://www.mdpi.com/2079-4991/12/11/1845

R: As requested by the reviewer, we have inserted new paragraphs addressing all of these general concerns, including aspects of the immune response, biocorona, toxicity, and standards for CNT manufacturing. The reviewer suggests presenting a critical viewpoint on the use of carbon nanomaterials as diagnostic platforms for the detection of biomarkers, including labile phosphopeptides and the use of composites with metals for affinity chromatography analysis. I agree with the reviewer's request, and now all these issues have been addressed. The related information has been inserted on pages 16 to 17, lines 533 to 555.

Reviewer 2 Report

This is a really interesting and well written review article on the current state of the field in the use of carbon based nanomaterials and aloe vera in combination for melanoma. Very interesting indeed, and will definitely be of interest to the readership of Pharmaceutics. The only comments I have are style based which are:

Figures - the chemical structures seem hand-drawn, please replace with appropriate chemdraw

Figure 2 is too large and spans over mutliple pages - please separate if possible into different figures so the reader can easily see what they are looking at.

Tables - would it be possible to put a separate column in your tables for the references, it would make it easier to follow than having them in the first column

Author Response

Reviewer #2

The reviewer mentions that:

- This is a really interesting and well written review article on the current state of the field in the use of carbon based nanomaterials and aloe vera in combination for melanoma. Very interesting indeed, and will definitely be of interest to the readership of Pharmaceutics. The only comments I have are style based which are:

- Figures - the chemical structures seem hand-drawn, please replace with appropriate chemdraw

R: As noted by the reviewer, all figures in the manuscript have been created in watercolor art individually, with strict criteria for their chemical structure-structural conformation of their ligands and compounds, all of which are based on validated scientific references, as shown in the citation after each image.

- Figure 2 is too large and spans over mutliple pages - please separate if possible into different figures so the reader can easily see what they are looking at.

R: As requested by the reviewer, Figure 2 (page 34 to 40, line 1306) is now reformatted in the "landscape" style so that it can be easily viewed and understood by the reader.

- Tables - would it be possible to put a separate column in your tables for the references, it would make it easier to follow than having them in the first column

R: As requested by the reviewer, the tbles have been reformatted with the first column deleted and by inserting the references right after the last comments in the final column of each table to make them easier to understanding and the references easier to identify (Table 01, page 4, line 147 and Table 2 page 7, line 202).

Reviewer 3 Report

The paper is an interesting review about the use of aloe vera as derivatizing agent for carbon-based nanomaterials for the fabrication of antineoplastic agent in the treatment of melanoma. The paper is of high interest and well organized and well written. It is an opinion of this reviewer that it should be published in Pharmaceutics. Only few comments should be addressed as listed below:

In the last section of the paper, authors point of views about balance between potentialities and limitations for clinical applications should be improved.

Author Response

Reviewer #3

The reviewer mentions that:

  • The paper is an interesting review about the use of aloe vera as derivatizing agent for carbon-based nanomaterials for the fabrication of antineoplastic agent in the treatment of melanoma. The paper is of high interest and well organized and well written. It is an opinion of this reviewer that it should be published in Pharmaceutics. Only few comments should be addressed as listed below:
  • In the last section of the paper, authors point of views about balance between potentialities and limitations for clinical applications should be improved.

R: I agree with the reviewer and have added new points about the potential applications and imitations of using these nanomaterials in the clinic to the last section of the paper, as noted on pages 23, lines 777 to 802.

Reviewer 4 Report

Title: Carbon nanomaterials and Aloe vera against melanomas - where are we? Recent updates

This is very comprehensive and promising piece of work. I find the paper valuable and I believe it can be published after solving following issues. I suggested only minor corrections, hoping that they can be arranged by the authors.

1. Line 58:  Instead 6nm should be written 6mm

2. Line 211: calcium, potassium, magnesium, zinc are not compounds, but the elements and should be listed separately, after all other mentioned compounds.

3. Line 257: Behind HaCaT should be added  cell line

4. lines 527/528: no need for words dog and cat

5. In vitro and in vivo should be written in italic throughout the text

6. Aloe vera in the title should be written in italic

7. One single sentence about fullerenol should be added in the part 4.2. Fullerenes, since C60 is mentioned in part 5. line 840/841

Author Response

Reviewer #4

The reviewer mentions that:

- This is very comprehensive and promising piece of work. I find the paper valuable and I believe it can be published after solving following issues. I suggested only minor corrections, hoping that they can be arranged by the authors.

  1. Line 58:  Instead 6nm should be written 6mm

R: I agree with the reviewer, and have made the requested changes (page 2, line 58).

  1. Line 211: calcium, potassium, magnesium, zinc are not compounds, but the elements and should be listed separately, after all other mentioned compounds.

R: According to the reviewer´s suggestion, we have made these requested changes by listing these compounds separately, as shown on page 6, lines 179 to 180.

  1. Line 257: Behind HaCaT should be added  cell line

R: As requested, we have inserted the cell line in the text on page 8, line 213.

  1. lines 527/528: no need for words dog and cat

R: I agree with the reviewer and have removed the words “dog” and “cat” from the text.

  1. In vitro and in vivo should be written in italic throughout the text

R: As requested, in vivo and in vitro have been rewritten in italics throughout the text.

  1. Aloe vera in the title should be written in italic

R: As requested, Aloe vera has been rewritten in italics in the title.

  1. One single sentence about fullerenol should be added in the part 4.2. Fullerenes, since C60 is mentioned in part 5. line 840/841

R: I agree with the reviewer, and as requested, a brief discussion of fullerenes has been added on pages 14 to 15, lines 471 to 475.
